# Fracture Toughness of Short Fibre-Reinforced Composites—In Vitro Study

**DOI:** 10.3390/ma17215368

**Published:** 2024-11-02

**Authors:** Noor Kamourieh, Maurice Faigenblum, Robert Blizard, Albert Leung, Peter Fine

**Affiliations:** UCL Eastman Dental Institute, London WC1E 6DE, UK; mjfay@btinternet.com (M.F.); r.blizard@ucl.ac.uk (R.B.); albertleung@rcsi.ie (A.L.); p.fine@ucl.ac.uk (P.F.)

**Keywords:** short fibre-reinforced composite, fracture toughness, *everX Posterior*, *everX Flow*, glass fibres, biomimetics

## Abstract

The development of dental materials needs to be supported with sound evidence. This in vitro study aimed to measure the fracture toughness of a short fibre-reinforced composite (sFRC), at differing thicknesses. In this study, 2 mm, 3 mm and 4 mm depths of sFRC were prepared. Using ISO4049, each preparation was tested to failure. A total of 60 samples were tested: 10 samples for each combination of sFRC and depth. Fractured samples were viewed, and outcomes were analysed. EXF showed greater toughness than EXP, with a mean of 2.49 (95%CI: 2.25, 2.73) MPa.m^1/2^ compared to a mean of 2.13 (95%CI: 1.95, 2.31) MPa.m^1/2^, (F(1,54) = 21.28; *p* < 0.001). This difference was particularly pronounced at 2 mm depths where the mean (95%CI) values were 2.72 (2.49, 2.95) for EXF and 1.90 (1.78, 2.02) for EXP (Interaction F(2,54) = 7.93; *p* < 0.001). Both materials performed similarly at the depths of 3 mm and 4 mm. The results for both materials were within the accepted fracture toughness values of dentine of 1.79–3.08 MPa.m^1/2^. Analysis showed crack deflection and bridging fibre behaviour. The optimal thickness at the cavity base for EXF was 2 mm and for EXP 4 mm. Crack deflection and bridging behaviour indicated that restorations incorporating sFRCs are not prone to catastrophic failure and confirmed that sFRCs have similar fracture toughness to dentine. sFRCs could be a suitable biomimetic material to replace dentine.

## 1. Introduction

All dental restorations will ultimately suffer deterioration and degradation in clinical service [1]; thus, there is a need to design dental materials with properties that will reduce the incidence of treatment failure [2,3]. Coupled with this is the need to understand and utilise the principles of minimally invasive dentistry (MID) [4], which focuses on preserving the foundation and structure of natural teeth. Research and development in dental restorative materials aim to address the challenge of which materials are most suited to restore the form and function of teeth [5], thus following the principles of Biomimetics [6,7]. Short fibre-reinforced composites (sFRCs) have been introduced to replicate dentine in the restoration of broken-down teeth, which demonstrates resilience to failure. Short fibre-reinforced composites are conventional resin composites into which electrical E-glass fibres are inserted [7] and in so doing are specifically designed to mimic the supporting role of fibres in dentine. They were first introduced in 2013 [8]. sFRCs are available in two forms: everX Posterior^TM^ (EXP, GC, Tokyo, Japan) and everX Flow^TM^ (EXF, GC, Tokyo, Japan) (see Table 1).

Dentine is an example of a naturally occurring fibre-reinforced tissue consisting of collagen fibres approximately 23 μm in length and 20–400 nm in diameter [9], within a hydroxyapatite and carbonated matrix, and as such is regarded as a fibre-reinforced naturally occurring material [10]. A direct comparison between everX Posterior and everX Flow was desirable in order to clinically decide which material was better in any given situation and to compare their properties with natural dentine.

The resin combination within sFRCs creates a semi-interpenetrating polymer network (semi-IPN) when polymerised and enhances the toughness of the material [7]. The physical properties of sFRCs have suggested they are is an appropriate material as a dentine replacement substructure in posterior teeth, covered by a layer of 1–2 mm conventional particulate-filled composites [11].

Masticatory forces on posterior teeth range from 8 to 880 Newtons (N) [12]. Demarco et al. [2] reported that with respect to restorations, ‘To prevent fractures the strongest materials should be used with high fracture toughness’ (K_IC_), which describes a material’s damage tolerance and resistance to catastrophic failure/crack propagation under an applied load [13,14]. A fracture toughness test (K_IC_) [15] measures the amount of energy required to cause a material to fracture, measuring stress intensity at the tip of a crack, from where the fracture propagates [16,17], and is measured as Mpa.m^1/2^ [17,18,19]. Fracture toughness testing is an important method of assessing a material’s ability to undergo stress without fracturing and survive internal crack propagation occurring prior to failure [15]. A constant displacement rate of 1.0 mm/min speed (crosshead speed) is commonly used [10,18,20,21,22,23,24,25,26]. As a material designed to replace dentine, it is important to relate the fracture toughness of sFRCs with that of natural human dentine.

**Table 1 materials-17-05368-t001:** Properties of two short fibre-reinforced composites—EXF and EXP.

sFRC	*everX Posterior* (EXP)	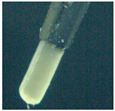	*everX Flow* (EXF)	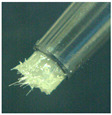
Launched globally	2013		2019	
Cure depth	4 mm		5.5 mm	
% of fibres (*w*/*w*)	E-glass fibres 5–15%		E-glass fibres 25%	
% of particle fillers (*w*/*w*)	Barium glass: 60–70% Silicon dioxide: 1–5%		Barium glass: 42–52% Silicon dioxide: Trace	
% of resin matrix (*w*/*w*)	Bis-GMA: 10–20% TEGDMA: 5–10%		Bis-MEPP 15–25% TEGDMA: 1–10% UDMA: 1–10%	
Fibre length	800 µm		140 µm	
Fibre diameter	17 µm		6 µm	
Indications	Dentine replacement in large posterior cavities			
	Endodontically treated teeth			
	Cavities with missing cusps			
			Dentine replacement in small cavities	
			Core build-up	
GC manufacturer [8], Lassila et al., 2020 [24]				

The recorded fracture toughness of dentine varies depending on the parameters of the testing methods. El Mowafy and Watts [27] demonstrated a dentine fracture toughness of 3.08 Mpa.m^1/2^ using a notched sample without a sharp crack. However, Imbeni, Nalla [28] used a 3-point bend test and demonstrated a dentine K_IC_ of 1.79 Mpa.m^1/2^ for samples prepared with both a notch and a sharp pre-crack (using a sharp razor blade) and 2.72 Mpa.m^1/2^ for samples prepared only with a notch. The fracture toughness of dentine is related to the orientation of dentinal tubules and mineralisation of the tissue [29].

As a material that is considered to mimic that of a natural tooth, it is important to understand sFRCs’ ability to resist fracture [15]. E-glass fibres inserted into composites alter the microstructure and are designed to enhance the mechanical properties of sFRCs, depending on fibre diameter, orientation, loading and length [23]. When the fibre lengths exceed 0.5–1.6 mm, the ‘critical fibre length’, the fibres are at the highest tensile strength and the matrix at the maximum shear strength [30]. Fibres in sFRCs exceed critical length, allowing for the transmission of stress from the semi-IPN resin matrix to E-glass fibres [26,31].

Table 1 outlines the details of sFRCs, *everX Posterior* (EXP) and *everX Flow* (EXF), consisting of discontinuous fibre-reinforced composites mimicking the fibrous structure of dentine [32]. The enhanced properties of sFRCs may be due to the ‘crack stopper’ action of the fibres preventing crack propagation [11,33,34].

sFRCs used as a base for directly layered restorations have been reported to significantly improve the restorations’ performance [35]. The fracture toughness (K_IC_) of dentine has been recorded as 3.1 Mpa.m^1/2^ [17]; by contrast, amalgam has a K_IC_ of 1.3–1.6 Mpa.m^1/2^, and a resin composite has one of 1.4–2.3 Mpa.m^1/2^ [7,15,17,36]. These existing studies have found the K_IC_ of sFRCs to mimic that of dentine [7,10,20], with some recording a mean K_IC_ of 2.8 Mpa.m^1/2^ [10] and 3.1 Mpa.m^1/2^ [26]. It would seem preferable to choose such biomimetic materials that mimic the characteristics of a natural tooth [7,37].

The reduction in filler particles in sFRCs due to space being occupied by fibres (Tsujimoto et al., 2016b [38]) leads to limitations in the polishability of sFRCs, necessitating their use as an internal build-up material below a conventional veneering material, (Magne and Belser, 2022 [39]). Nonetheless, K_IC_ is an essential factor to consider when considering biomimetic replacement for tooth tissue [10].

This study aimed to assess the effect of varying the depths of sFRCs on fracture toughness during in vitro testing and their capability to mimic dentine.

## 2. Materials and Methods

The fracture toughness testing of everX posterior and everX Flow was undertaken to compare which of these new materials were most suited to restoring posterior teeth at varying depths. This will potentially have clinical implications when deciding what material to use in specific clinical situations. Two materials were chosen to test the fracture toughness of sFRCs: (i) EXP and (ii) EXF (see Table 1). The preparation of the samples was in accordance with International Organization for Standardization ISO4049 [40] with modifications incorporated into the test sample size and the methods of creating a pre-crack in the samples.

A rapid review of the toughness of sFRCs was carried out. In spite of substantial heterogeneity, a pooled estimate of 2.62 (SD 0.45) MPa.m^1/2^ was calculated. A group sample size of ten provided a precision (95%CI) of ±0.28 MPa.m^1/2^ (Unpublished MSc project, 2023). For comparison between EXP and EXF, a sample size of 30 provided 80% power at 5% significance to detect a difference of 0.11 MPa.m^1/2^.

In the context of this study, a ‘small cavity’ was considered to have a depth of 3–4 mm, a ‘medium cavity’ 4–5 mm and a ‘large cavity’ 5–6 mm. The depth of the overlying veneering material (replacing enamel) would be 1–2 mm. Test samples were prepared in Polytetrafluoroethylene tape (PTFE) moulds with rectangular cut outs sized 2 mm × 2 mm × 25 mm, 3 mm × 2 mm × 25 mm and 4 mm × 2 mm × 25 mm (see Figure 1).

Figure 1 illustrates the moulds used to develop the samples. The three depths are all present on the same structure. Each slot was filled with either EXF or EXP to the appropriate depth.

These varying thicknesses of the test material were designed to replicate the depth of a dentine replacement material in typical cavities in posterior teeth. The samples were cured using a 9 overlapping irradiation window technique ISO4049 [40]. An LED curing light 700 mW/cm^2^ (Model DEMI ^PLUS^ Kerr Corporation, Orange, CA, USA) was used for 20 s per window, as per the manufacturer’s recommendations [8,41]. A natural sharp pre-crack, as close to 0.1 mm as possible, was introduced into each sample using a razor blade ISO23146 [42]. To confirm the crack depths of each sample, a 3D microscope (Tucsen Photonics Co., Ltd., Fuzhou, Fujian, China) was used to measure the crack depth (60 samples). Each sample was tested using a 3-point bend test jig (see Figure 2) within the universal testing machine (Shimadzu, Model AGS-X Shimadzu Corp., Kyoto, Japan) at a crosshead speed of 1.0 mm/min until specimen fracture (see Figure 3). The force required to fracture was recorded in Newtons, (N), to provide data to calculate fracture toughness (MPa.m^1/2^) (see Table 2). Following fracture toughness testing, each sample was assessed under a scanning electronic microscope (SEM) (Ziess Sigma 300VP Field Emission Scanning Electron Microscope [Carl Zeiss Ltd., Cambourne, UK]), to investigate how the fractures propagated following the catastrophic failure of the sample.

Data were entered into an IBM SPSS (IBM Corp. Released 2021. IBM SPSS Statistics for Windows, Version 28.0. Armonk, NY, USA: IBM Corp) spreadsheet. The results are presented as means, standard deviations (SDs) and 95% Confidence Intervals (95%CIs). Group comparisons were carried out by a two-way Analysis of Variance (ANOVA) procedure.

## 3. Results

Table 3 illustrates the mean fracture toughness of EXF and EXP at varying depths. EXF exhibited a mean fracture toughness of 2.49 MPa.m^1/2^ (SD 0.39 MPa.m^1/2^) compared to a mean of 2.13 MPa.m^1/2^ for EXP (SD 0.29 MPa.m^1/2^). The mean fracture toughness for EXF and EXP with error bars indicated no overlap at 2 mm, signifying that at 2 mm, EXF and EXP were significantly different, with EXF achieving a higher fracture toughness than EXP of 2.72 MPa.m^1/2^ (0.37) (see Figure 4).

The ANOVA revealed a statistically significant difference in fracture toughness between the materials. EXF exhibited greater toughness than EXP, with a mean of 2.49 (95%CI: 1.95, 2.31) MPa.m^1/2^, compared to a mean of 2.13 (95%CI: 01.95, 2.31 MPa.m^1/2^, (F(1,54) = 21.28; *p* < 0.001). This difference was particularly pronounced at the smallest block size (2 mm × 2 mm × 25 mm) where the mean (95%CI) values were 2.72 (1.95, 2.31) for EXF and 1.90 (1.78, 2.02) for EXP (Interaction F(2,54) = 7.93; *p* < 0.001). Both preparations performed similarly at the depths of 3 mm and 4 mm. The results for both materials were within the recognised/accepted fracture toughness values of dentine of 1.79–3.08 MPa.m^1/2^. The standard deviations (Table 3) also showed that for all depths, the results for EXF were more variable than those for EXP.

Assessing the fractured samples following testing to the point of fracture demonstrated that all samples remained tethered (attached), except for five samples of EXF 4 mm which recorded catastrophic failure (fractured in half) (see Figure 5c). Figure 5 also shows fibre protrusion in some samples that had wider cracks (image EXP 4 mm (e and f)). Image EXF 4 mm (c) illustrates the internal fractured surface of one of the samples that broke catastrophically. EXF 4 mm (b) and EXP 4 mm (e and f) demonstrated how 4 mm specimens bent during testing. The fracture pattern on EXP 2 mm was straighter and less well defined (see Figure 5d). The toughness of EXF recorded a mean value of 2.49 MPa.m^1/2^ (0.39 MPa.m^1/2^), whilst that of EXP was 2.13 MPa.m^1/2^ (0.29 MPa.m^1/2^).

SEM imaging and analysis revealed that the extent to which the cracks travelled down the remaining samples varied, with the majority not reaching the opposing surface (see Figure 6a). In the EXF 2 mm group, none achieved a crack that extended to the opposing surface, and only one sample recorded a crack that extended to within 500 µm of the opposing surface. EXF presented a more irregular chaotic and zig-zagging fracture pattern (see Figure 6a,b) compared to the straighter appearance of EXP (see Figure 6c). With EXF samples, more irregular cracks achieved a higher fracture toughness (Mpa.m^1/2^).

SEM imaging allowed for measurements of fibre diameter for EXF and EXP. Figure 7 shows the difference in fibre diameter and correlates with the figures from the manufacture in Table 1, as well as demonstrating that EXF samples had short and randomly distributed fibres, as shown in Figure 8.

As seen in Figure 6 and Figure 9, SEM analysis provided information on the fibre behaviour following fracture toughness testing and showed that fibres stopped the crack’s path of propagation and created a zig-zag pattern where the fibre interrupted the path of fracture. Fibres demonstrated both a bridging behaviour (see Figure 9a,b) as well as ‘pull out’ (see Figure 9e,f) and indications of potential fibre fracture (see Figure 9c,d).

## 4. Discussion

This study demonstrated that both EXP and EXF achieved a fracture toughness similar to that of dentine as reported by El Mowafy and Watts [27] and Imbeni, Nalla [28] within the current study’s accepted range for dentine fracture toughness of 3.08 MPa.m^1/2^–1.79 MPa.m^1/2^.

Crack propagation in this study was initiated by creating a minimal (~0.1 mm) pre-crack; this is in keeping with ISO23146 [42], using a razor blade to create a consistent, natural, sharp, central pre-crack. Other pre-crack techniques were considered but discounted as the aim was to produce a fracture that would replicate a naturally occurring clinical fracture. Other studies used a notch ISO4049 [40] to guide the pre-crack; however, this was deemed unnecessary for limiting the depth alteration for each sample.

The results of the current study agreed with those by Bijelic-Donova, Garoushi [20] and Lassila, Sailynoja [10] who found that the fracture toughness value of sFRCs matched that of dentine. Both materials can be used to an appropriate standard. However, EXF seems preferable to use in a smaller thickness, i.e., 2 mm, and therefore is the better choice in shallower cavities. This is in agreement with the manufacturer’s (2020) recommendations. EXP and EXF behave similarly in larger thicknesses such as 4 mm. This suggests that in larger cavities, the ease of use and clinician’s preference in individual clinical situations should determine whether flowable EXF or packable EXP is preferred. Previous literature reviews/meta-analyses estimated a mean of 2.62 MPa.m^1/2^ for sFRCs, and in this study, similar results were demonstrated, as well as EXF 2 mm achieving a higher K_IC_ of 2.72 MPa.m^1/2^.

Garoushi et al. [18] suggested that the fibres’ bridging behaviour and dissipating energy slowed crack propagation and prevented catastrophic failure, exposing the fibre ends at the fracture surface. Studying the samples under SEM demonstrated glass fibre orientation and behaviours. The fibre behaviours shown in Figure 9 demonstrated similar findings to those by Abdul-Monem, El-Gayar, Al-Abbassy [44], Abouelleil et al. [45], Alshabib, Silikas, Watts [46], Bijelic-Donova et al. [20], Huang et al. [22], Lassila et al. [23], Tsujimoto et al. [26] and Tsujimoto et al. [38]. Because of the recorded fibre behaviour, sFRCs exhibited a more natural and more ‘graceful’ fracture, where fibre ends protrude at the fracture surface, demonstrating ‘fibre pull out’ rather than catastrophic failure [10,13]. However, *everX Flow* (EXF) did show a superior fracture toughness and could be an improved version with smaller fibres improving the ease of use [39]. From a clinician’s perspective, it is helpful to understand that these relatively new materials have remained consistent throughout their development.

EXF samples had shorter more randomly distributed fibres that pulled out more easily. EXP, which has longer fibres that are less likely to be randomly orientated, demonstrated a straighter fracture path. EXF has an irregular crack pattern where the crack’s path was disrupted by coming into contact with fibres and deviated in its path/direction, thus prolonging the propagation of the crack. The fact that only one sample of EXF 2 mm recorded a crack within 500 µm of the opposing surface indicates that it is a material that withstood the fracture toughness testing and therefore should be considered suitable for its designated purpose. Fracturing EXF 2 mm required more force (N) and achieved a higher fracture toughness (Mpa.m^1/2^). Thus, tougher materials have more irregular cracks.

Within this study, the randomly orientated E-glass fibres within sFRCs were identified as resisting crack propagation. Garoushi, Sailynoja [31] recorded them acting as ‘crack stoppers’ and ‘crack deflectors’, thus improving the fracture toughness of the material. However, some other studies indicated that perhaps the fracture toughness is a function not just of the glass fibres in sFRCs but also a function of the semi-IPN resin matrix [26,31,44]. However, further research would aid in understanding the material and its ability to mimic the K_IC_ of dentine and the clinical situations that best suit the materials’ properties.

sFRCs have reduced polishability due to the reduction in filler particles to allow space to be occupied by fibres. Magne and Milani [35] investigated the use of sFRCs as a liner below a CAD/CAM indirect restoration which gave them superior mechanical properties. Combined with this study’s finding of EXF 2 mm providing the most superior fracture toughness, it may be considered that 2 mm EXF as a base below an indirect restoration would be favourable. The current investigation indicated that there is potential for less catastrophic failures in sFRCs: restorations. This is supported by Garoushi, Sungur [34] who suggested the possibility of more ‘repairable failure’ in restorations that utilised EXF as a base for both direct and indirect restorations. In addition, there is greater potential for more repairable failures [10,13,34]. Therefore, it should be considered a suitable and biomimetic material to replace dentine when restoring teeth.

### Limitations

Time and armamentarium available were limiting factors to this study. In addition, the use of human tissue, i.e., dentine, was unavailable and would have required more stringent ethical approval. The obvious limitations of an in vitro study make these findings not directly transferable to the in vivo environment.

## 5. Conclusions

This study suggests that sFRCs have similar fracture toughness to natural dentine and are therefore a suitable dentine replacement material. However, clinicians need to be aware of the cavity depth when selecting the most appropriate form of sFRC. In a posterior restoration requiring 2 mm of dentine replacement, EXF is the preferred material. At depths of 3–4 mm, either EXF or EXP can be used. Further investigations of the material can provide a better understanding as can evidence-based decisions for clinicians when considering the incorporation of sFRCs into their clinical practice. The ability to provide a biomimetic restoration reduces the incidence of catastrophic fractures of restorations and the resulting complications.

## Figures and Tables

**Figure 1 materials-17-05368-f001:**
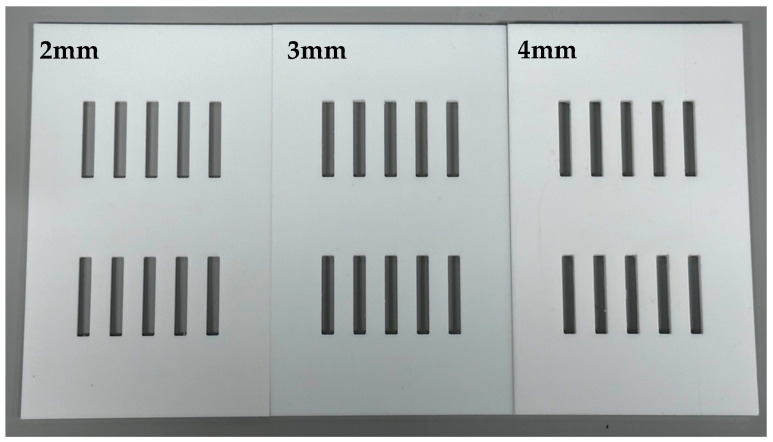
PTFE moulds made from PTFE sheets with 2 mm, 3 mm and 4 mm depth CAD CAM-designed rectangular cut outs.

**Figure 2 materials-17-05368-f002:**
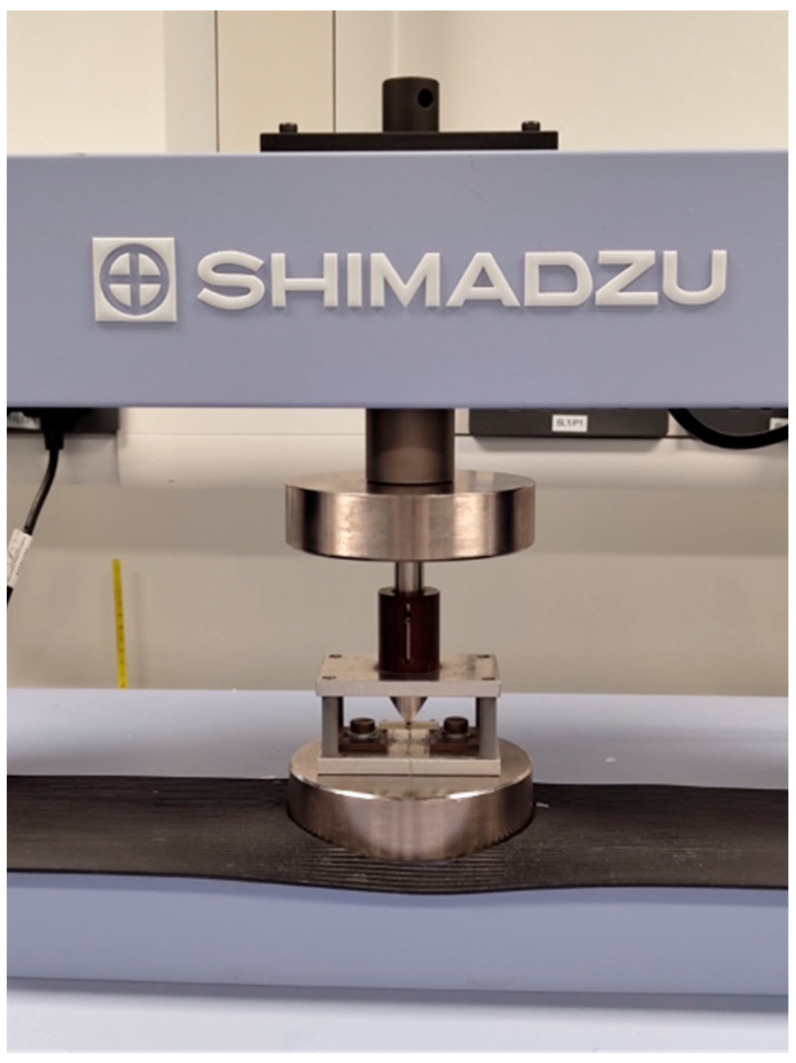
Shimadzu universal testing machine with a sample loaded in the test rig—3-point bend test jig.

**Figure 3 materials-17-05368-f003:**
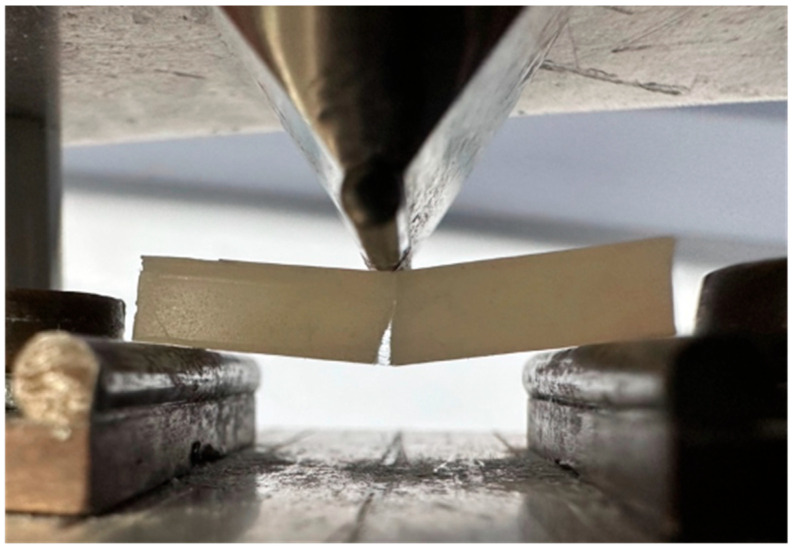
An example of a 4 mm EXP fractured sample within the 3-point bend test jig after fracture toughness testing showing the protruding fibres within the crack.

**Figure 4 materials-17-05368-f004:**
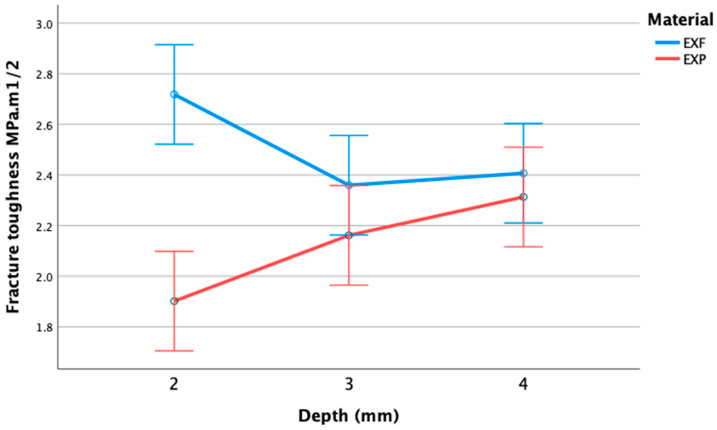
Mean fracture toughness of sFRCs EXF and EXP and depths with 95% CI.

**Figure 5 materials-17-05368-f005:**
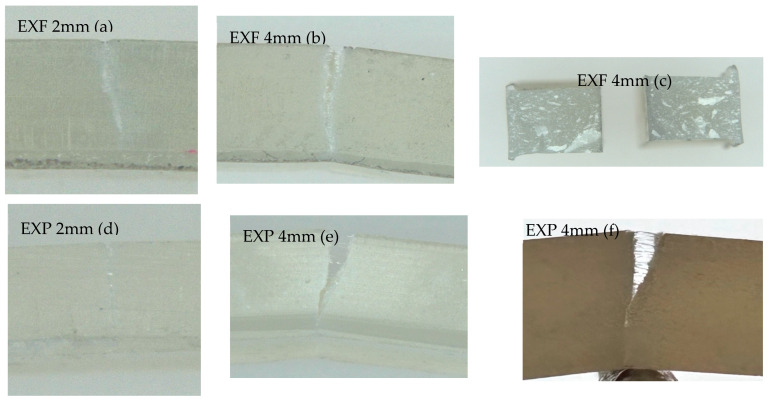
Fractured samples of EXF 2 mm and 4 mm and EXP 2 mm and 4 mm. Image (**a**) is EXF 2 mm after fracture; image (**b**) is EXF 4 mm after fracture; image (**c**) EXF 4 mm after catastrophic fracture; image (**d**) is EXP 2 mm after fracture; image (**e**,**f**) are EXP 4 mm after fracture.

**Figure 6 materials-17-05368-f006:**
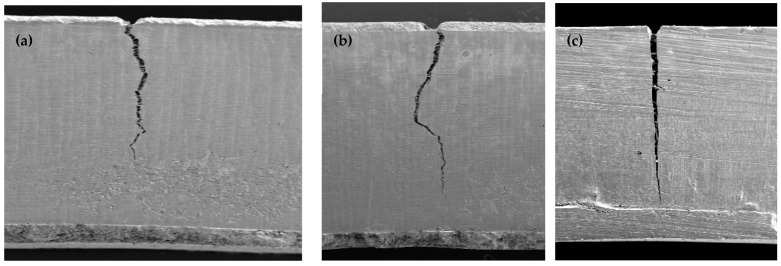
SEM image of crack propagation in (**a**) EXF 2 mm sample, (**b**) EXF 2 mm sample and (**c**) EXP 2 mm sample.

**Figure 7 materials-17-05368-f007:**
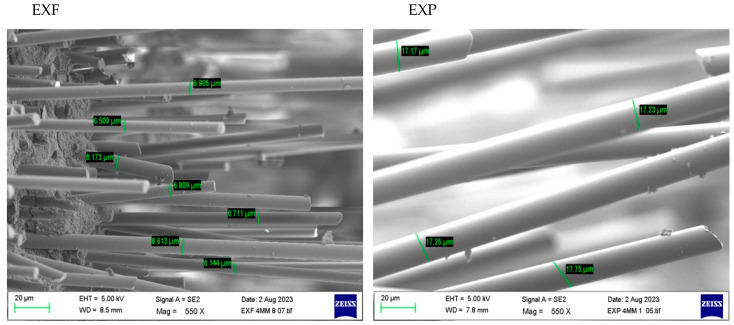
SEM of EXF and EXP fractured fibre diameters.

**Figure 8 materials-17-05368-f008:**
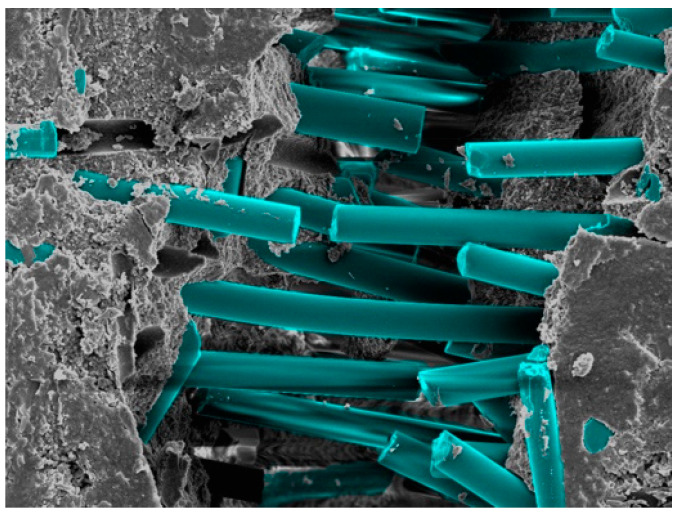
Colourised SEM image demonstrating random distribution of fibres in EXF.

**Figure 9 materials-17-05368-f009:**
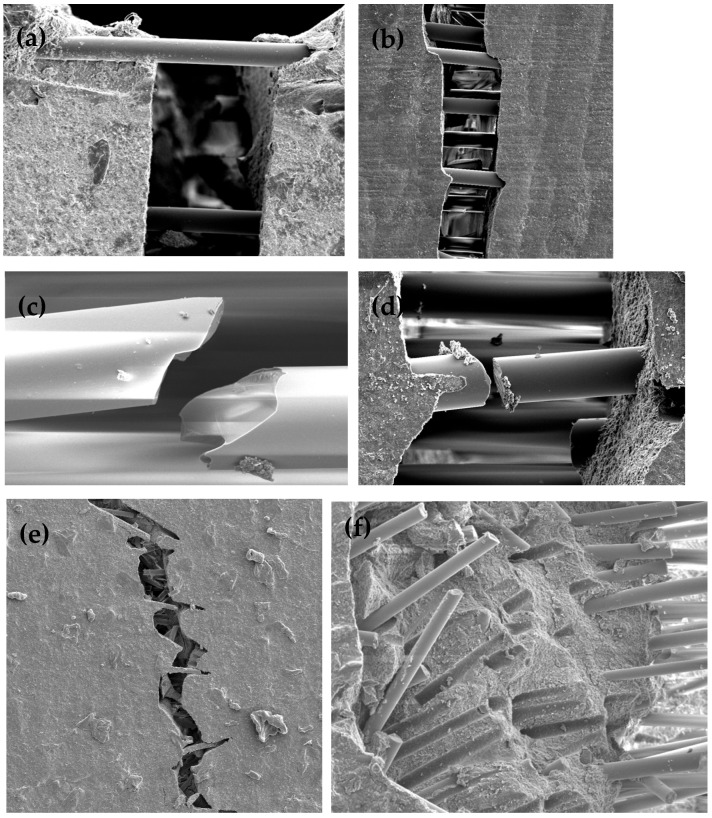
SEM images of fibre behaviour—bridging fibres, fractured fibres and fibre pull out. (**a**,**b**) EXP samples demonstrate fibre bridging, (**c**,**d**) EXP samples demonstrating potential fibre fracture, (**e**,**f**) EXF samples demonstrating fracture pull out.

**Table 2 materials-17-05368-t002:** The fracture toughness equation.

$K_{IC}$ = Fracture toughness	P_max_ = max load exerted on the specimen at fracture in Newtons (N)	L = length span	*b* = width	*h* = height	*a* = crack length
$K_{IC}=\frac{fP_{max}L}{{bh}^{3/2}}\sqrt{10^{-3}}\;mPa\;m^{1/2}$					
where					
$x=\frac{a}{h}$					
and f = the geometrical function dependent on x					
$f\left(x\right)={3x}^{1/2}\left[1,99-x\left(1-x\right)\left(2,15-3.93+2.7x^{2}\right)\right]/\left[2\left(1+2x\right)(1-{x)}^{3/2}\right]$					
ISO20795-1 [43]					

**Table 3 materials-17-05368-t003:** Means (MPa.m^1/2^) and standard deviations (MPa.m^1/2^) of fracture toughness of short fibre-reinforced composites EXP and EXF and depths.

	Depth (mm)			
Material	2	3	4	Material Mean
*everX Flow* (EXF)	2.72 (0.37)	2.36 (0.37)	2.41 (0.36)	2.49 (0.39)
	n = 10	n = 10	n = 10	n = 30
*everX Posterior* (EXP)	1.90 (0.19)	2.16 (0.29)	2.31 (0.23)	2.13 (0.29)
	n = 10	n = 10	n = 10	n = 30

## Data Availability

The raw data supporting the conclusions of this article will be made available by the authors on request.

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
