# Peer review of "Fracture Toughness of Short Fibre-Reinforced Composites—In Vitro Study"

_materials, 2024, doi:10.3390/ma17215368_

Round 1
Reviewer 1 Report
Comments and Suggestions for Authors
I would appreciate a correct description of the materials used, such as having international registration.
The summary does not reflect the description made of materials and methods; I would appreciate correction.
The testing machines, microscopes, photopolymerizers, etc. used must be well described and referenced.
Groups must be well described and correctly identified.
Show how the matrices were made and review the description of the execution methodology.
How many samples were used as controls?
Add photographs of the test procedure to enlighten the reader.
Attach a photograph of the jig used in the 3 point bend test.
Refer to the speed at which the 3 point bend test is carried out.
In the conclusions, remove bibliographic references, the text must be written by the authors where they validate their results in comparison with others. You should do it in the discussion.
Author Response
Reviewer 1:
Thank you for your helpful comments which we have reviewed.
- With respect to the description of the materials used, we have included an overall description of short fibre reinforced composite in both its forms in the introduction and have now expanded on this in the introduction and methods & materials sections.
- We are aware of the limitations of the prescribed word count and feel that our current explanation and the additional information that we have supplied is adequately reflected in the summary. We have also made changes to the design of Table 1 to clarify the measurements.
- The descriptions of the testing machines have been revised and referenced.
- With respect to your comment about needing better description of the groups, we have explained the different materials in greater depth but have not referred to the tests as ‘groups’.
- We have now included a description of how the matrices were made, including an image of the test block.
- There are no controls in this experiment. The experiment compares two materials at 3 thicknesses.
- We have added photographs of the test procedure
- We have added a photograph of the jig used
- We have made reference to the cross head speed at which the 3 point bend test was carried out.
- We have made the corrections to the conclusion as requested.
Reviewer 2 Report
Comments and Suggestions for Authors
This in-vitro study aimed to compare the fracture toughness of two types of short fibre reinforced composite (sFRC), at different thicknesses, in order to offer to the clinicians a more suitable and biomimetic material to replace dentine when restoring decayed teeth, a material with a similar fracture toughness in order reduces the incidence of catastrophic fractures of the restorations.
The manuscript fit the journal scope, is clear, and can be relevant for the field and I appreciate the quality of the images, but in my oppinion some modifications should be done:
1. The abstract should be a total of about 200 words maximum and should be a single paragraph (according to the Instructions for Authors). The names of the two types of materials used in this study should be mentioned, not just their abbreviation.
2. The Introduction it is too long. It should briefly place the study in a broad context and highlight why it is important. It should define the purpose of the work and its significance, including specific hypotheses being tested. In this manuscript the Introduction contains a lot of information that are more suitable for Materials and Method and even for the Discussions section. One exemple: the paragraph about EXF and EXP properties and the Table 1 should be moved to the Materials and Methods, because these are the two materials used in the study. My advice is to rewrite this section, in order to improve the clarity. Keep the introduction comprehensible to scientists working outside the topic of the paper.
3. Materials and Method should be described with sufficient details to allow another researcher to replicate and build on published results
- all the details about the EXF and EXP, including Table 1. should be moved from the Introduction.
- the description of the samples should be more clear
- what is PTFE abbreviation? Maybe not all the journal readers are familiar with this....
- there are missing the names and producers of some devices and tool used in this experiment : LED curing light (?…), a 3D microscope was used (…?.), universal testing machine (Shimadzu – it is not enough..), a scanning electronic microscope (SEM)(?…). Materials and Method should be described with sufficient detail to allow others to replicate and build on published results
4. The Conclusion section should not contain any citation.
5. The cited references are relevant but more than half of them are not recent publications (within the last 5 years, as mentioned in the Instructions for Authors)
Comments on the Quality of English Language
This in-vitro study aimed to compare the fracture toughness of two types of short fibre reinforced composite (sFRC), at different thicknesses, in order to offer to the clinicians a more suitable and biomimetic material to replace dentine when restoring decayed teeth, a material with a similar fracture toughness in order reduces the incidence of catastrophic fractures of the restorations.
The manuscript fit the journal scope, is clear, and can be relevant for the field and I appreciate the quality of the images, but in my oppinion some modifications should be done:
1. The abstract should be a total of about 200 words maximum and should be a single paragraph (according to the Instructions for Authors). The names of the two types of materials used in this study should be mentioned, not just their abbreviation.
2. The Introduction it is too long. It should briefly place the study in a broad context and highlight why it is important. It should define the purpose of the work and its significance, including specific hypotheses being tested. In this manuscript the Introduction contains a lot of information that are more suitable for Materials and Method and even for the Discussions section. One exemple: the paragraph about EXF and EXP properties and the Table 1 should be moved to the Materials and Methods, because these are the two materials used in the study. My advice is to rewrite this section, in order to improve the clarity. Keep the introduction comprehensible to scientists working outside the topic of the paper.
3. Materials and Method should be described with sufficient details to allow another researcher to replicate and build on published results
- all the details about the EXF and EXP, including Table 1. should be moved from the Introduction.
- the description of the samples should be more clear
- what is PTFE abbreviation? Maybe not all the journal readers are familiar with this....
- there are missing the names and producers of some devices and tool used in this experiment : LED curing light (?…), a 3D microscope was used (…?.), universal testing machine (Shimadzu – it is not enough..), a scanning electronic microscope (SEM)(?…). Materials and Method should be described with sufficient detail to allow others to replicate and build on published results
4. The Conclusion section should not contain any citation.
5. The cited references are relevant but more than half of them are not recent publications (within the last 5 years, as mentioned in the Instructions for Authors)
Author Response
Reviewer 2:
It is unclear what reviewer 2 means by: ” Extensive editing of English Language is required”. For all authors English their first language and despite repeatedly re-reading the manuscript, we are uncertain what changes to the English language could be made. As the other two reviewers have not made similar comments, we have decided not to address this perceived issue other than clarifying points that needed to be clarified.
- Thank you for pointing out that our abstract was too long. This has been rectified.
- The introduction has also been shortened as requested. We feel that some detail within the introduction is needed so that the lay reader has a clear understanding of what this study is looking at. Thank you for your suggestion that Table 1 and the paragraph about EXP and EXF should be incorporated in the Methods & Materials; this has been done.
- In the materials and methods, we have written the full wording for the abbreviation PTFE.
- We have made the description of the samples clearer.
- We have extended the explanation of the devices used, including references to their manufacturers. Hopefully now anyone wishing to replicate this study will be able to do so.
- Citations have been removed from the conclusion.
- We acknowledge that some of the references used in this manuscript are historical; this is to illustrate how the development of these materials has occurred and are therefore considered to help the authors tell the story.
Reviewer 3 Report
Comments and Suggestions for Authors
Fracture Toughness of Short Fibre Reinforced Composite—An In-Vitro Study."
This in-vitro study seeks to assess the fracture toughness of two types of short fibre reinforced composite (sFRC) at varying thicknesses.
Abstract: The abstract would benefit from a clearer explanation of the statistical methods employed, along with an indication of the statistical significance and relevance of the findings.
Introduction: The introduction should provide a more detailed rationale for the specific comparison of EXF versus EXP. Clarifying the importance of this comparison would enhance the context of the study.
Materials and Methods: This section is currently too brief. It is essential to include a more comprehensive step-by-step description of the procedures to enable replication of the study by others. Additionally, more detailed descriptions of the SEM analysis process and the statistical methods used for data analysis should be provided. The rationale behind the selection of varying thicknesses also requires greater clarity.
Results: While the results are presented clearly, they could be further strengthened by including additional statistical measures such as confidence intervals and effect sizes, where relevant.
Discussion: The discussion should expand upon the practical clinical implications of the findings. Specifically, it would be helpful to address how the results may impact real-world applications and whether EXF can be reliably employed in a range of clinical scenarios, or if its performance is influenced by specific conditions.
Conclusion: The conclusion needs to be rewritten for greater clarity, as it currently lacks coherence and may confuse the reader.
Author Response
Reviewer 3:
Thank you for your comments and suggestions, which we taken on board.
- By reducing the word count of the abstract as requested by Reviewer 2, we hope that the explanation of the statistical method is clearer.
- We have provided more detail in the introduction to specifically compare the two materials and enhance the study. The rationale behind comparing the two materials has been expanded and explained.
- We have expanded the Materials & Methods section to include more information about SEM analysis and statistical methods.
- Within the results section we have included further statistical measures including confidence intervals etc. We have added 95%CI values where appropriate and for clarity we included a Data Analysis paragraph in the Materials & Methods section.
- We have expanded the discussion to include practical/clinical applications of the materials.
Round 2
Reviewer 1 Report
Comments and Suggestions for Authors
Congratulations on the work and thanks for the changes
Author Response
We are grateful to reviewer 1 for their time and helpful comments and delighted that you are now happy with the manuscript.
Many thanks
Reviewer 3 Report
Comments and Suggestions for Authors
One of the points that could weaken the results of this study is the absence of any control group. In in vitro studies like this one, a control group is highly advisable to compare more appropriately, especially if one wants to prove the superiority of some other materials. In this proposal, two types of short fiber-reinforced composite materials were compared, EXF and EXP; however, no control was made using conventional materials without reinforcement, such as conventional composites or non-reinforced resins, which would make it difficult to have a full idea about the advantages of the studied material.
Author Response
We would like to thank Reviewer 3 for their further comments, in particular about the use or non-use of a control group in this research.
1) The study that we had conducted was completely valid and sound and scientifically and robust in it detailed methodology. The need for a control group was not considered to be necessary or desirable as this would have added a further degree of complication to the study, and would have yielded very little advantage.
2) The focus of the research was to investigate the strengths of two types of short fibre reinforced composite and not to compare sFRC with a unreinforced traditional composite acting as a control. By doing so we were able to make clinical suggestions for the use of these materials and their suitability as a restorative material in various depths of cavities.